# HiSin: A Sinogram-Aware Framework for Efficient High-Resolution Inpainting

## Abstract

High-resolution sinogram inpainting is essential for computed tomography reconstruction, as missing high-frequency projections can lead to visible artifacts and diagnostic errors. Diffusion models are well-suited for this task due to their robustness and detail-preserving capabilities, but their application to high-resolution inputs is limited by excessive memory and computational demands. To address this limitation, we propose HiSin, a novel diffusion-based framework for efficient sinogram inpainting that exploits spectral sparsity and structural heterogeneity of projection data. It progressively extracts global structure at low resolution and defers high-resolution inference to small patches, enabling memory-efficient inpainting. Considering the structural features of sinograms, we incorporate frequency-aware patch skipping and structure-adaptive step allocation to reduce redundant computation. Experimental results show that HiSin reduces peak memory usage by up to 30.81% and inference time by up to 17.58% than the state-of-the-art framework, and maintains inpainting accuracy across.

## 1 Introduction

A sinogram is a 2D projection-domain representation of computed tomography (CT) data, where each row corresponds to a different acquisition angle and each column to a detector position. However, in both industrial and medical settings, acquiring full sinograms is often impractical due to radiation exposure risks or scanning time constraints. Thus, sinogram inpainting plays a vital role in CT reconstruction, enabling recovery of projection data lost due to sensor malfunction, limited-angle acquisition, or radiation dose reduction (Kalender, 2011). High-resolution sinograms, such as 2048×2048 or larger, are increasingly common in both synchrotron tomography and modern CT systems. In these settings, high spatial resolution is essential for capturing fine structural details and ensuring diagnostic reliability, and reducing resolution can directly compromise reconstruction quality and downstream analysis.

Diffusion models have demonstrated strong performance in image inpainting tasks due to their generative flexibility and ability to synthesize fine-grained structure (Ho et al., 2020; Sohl-Dickstein et al., 2015; Ho et al., 2020; Lugmayr et al., 2022; Saharia et al., 2022). However, when scaled to high-resolution inputs, their iterative denoising over full-resolution spaces leads to high memory and computational overheads. For example, at 2048×2048 resolution, some diffusion-based inpainting models may require more than 40 GB of GPU memory and several seconds of inference per image. In real-world CT pipelines, inference is typically performed on one image at a time rather than in large batches. However, even in this single-image setting, the memory footprint of diffusion models can still exceed the capacity of common hardware. Although end-to-end CT reconstruction models have recently been explored, projection-domain inpainting remains valuable because it is directly compatible with established pipelines and provides measurement-level corrections. These constraints are further exacerbated by deployment limitations. For example, industrial CT systems are often embedded in compact workstations with strict power and cooling budgets. Large hospitals or cloud platforms also cannot dedicate high-end GPUs exclusively to a single inference task, making memory efficiency a practical necessity. A number of methods have been proposed to improve diffusion inference. Step distillation techniques (Salimans & Ho, 2022; Meng et al., 2023) reduce the number of denoising steps required for generation. Architecture-focused approaches (Li et al., 2023; Zhang et al., 2024b; Zhu et al., 2024) compress the model via architectural modifications or pruning. Trajectory-level optimization methods (Lu et al., 2022) focus on optimizing the

sampling trajectory itself. While effective in RGB image domains, these methods do not address the structural challenges of sinogram data. They typically assume semantic continuity, uniform texture complexity, or fixed inference depth, which do not hold in the projection domain.

Sinograms are physical measurements governed by acquisition geometry. They contain highly directional structures, large smooth background regions, and frequency sparsity concentrated along specific angular bands. Their patches remain physically meaningful when isolated–they correspond to narrower angular coverage or limited detector ranges, but still represent valid projections (Slaney & Kak, 1988). In contrast, natural image patches may lose semantic context (e.g., cropping only an eye from a face image), making patch-wise inference less suitable (Zhang et al., 2023a).

These domain differences motivate a rethinking of how diffusion inference should operate on sinograms. To this end, we propose HiSin, a high-resolution sinogram inpainting framework built around three design principles: hierarchical structure modeling, frequency-aware computation allocation, and structure-adaptive inference depth. First, we introduce a resolution-guided progressive inference scheme. Instead of processing the full-resolution image from the start, we perform denoising across a resolution hierarchy–beginning with a low-resolution version of the sinogram to extract global structure. This global prior is then used to guide patch-wise inference at higher resolutions. According to this progressive inference, we avoid full-frame activation and significantly reduce peak memory usage. Besides, unlike prior methods that rely on global embeddings, condition tokens, or architectural changes to maintain global context (Avrahami et al., 2022; Zhang et al., 2023a; Tumanyan et al., 2023; Zhang et al., 2023b), our approach requires no retraining and operates entirely at inference time.

Second, sinogram data contain large background regions that are structurally simple and spectrally sparse (Slaney & Kak, 1988). To avoid redundant computation in such areas, we introduce a frequency-aware patch-skipping mechanism that filters out low-energy patches using a simple FFT-based score. These patches are bypassed during inference and approximated using representations, reducing computation without affecting inpainting fidelity. For the remaining patches, local complexity varies significantly—-from flat gradients to fine structural edges. To better align inference cost with content richness, we apply a structure-adaptive denoising scheduler, which assigns each patch a custom number of denoising steps based on a complexity score combining Shannon entropy (Shannon, 1948) and frequency energy. This joint design eliminates redundant computation in low-information regions and allocates denoising effort to structurally rich areas, thereby improving efficiency while preserving inpainting quality.

Together, these designs preserve inpainting fidelity by first capturing global structure that maintains long-range consistency across resolution levels. Patch skipping removes only regions with negligible signal, ensuring that no structurally relevant information is lost, while adaptive denoising assigns each patch a sufficient number of steps based on its complexity, preventing under-computation in detail-rich areas. As a result, HiSin improves efficiency (both in peak memory usage and inference speed) without compromising inpainting quality. In summary, our contributions are as follows:

- We propose HiSin, a sinogram-aware diffusion framework for high-resolution sinogram inpainting under strict memory constraints. By progressively extracting global structure at low resolution and refining patches at high resolution, it avoids full-frame activation, maintains long-range consistency, and requires no retraining or architectural changes.
- We introduce two inference-time mechanisms, frequency-aware patch skipping and structure-adaptive step allocation, which exploit the spectral sparsity and structural heterogeneity of sinogram data. These reduce redundant computation in trivial regions and align inference cost with local complexity.
- HiSin successfully enables 2048×2048 sinogram inpainting on an A100 GPU, reducing peak memory by up to 30.81% and inference time by up to 17.58% without compromising inpainting quality across multiple datasets, input resolutions, mask types, and mask ratios.

## 2 RELATED WORK

**Inpainting Methods for RGB and Projection-domain Images.** Diffusion-based models have achieved strong performance on image inpainting due to their generative capacity and ability to model uncertainty. Methods such as RePaint (Lugmayr et al., 2022), Palette (Saharia et al., 2022),

Blended Diffusion (Avrahami et al., 2022), and CoPaint (Zhang et al., 2023a) enhance global coherence on RGB images through iterative sampling or semantic priors. However, these methods assume spatially dense textures and uniform semantics, which do not align with the structural properties of projection-domain data such as sinograms. Sinogram inpainting has also been studied in industrial and medical CT settings using CNNs, GANs, transformer-based and diffusion-based models for sparse-view recovery or artifact reduction (Lee et al., 2018; Jin et al., 2017), but these works generally ignore memory constraints and inference time optimizations.

**Inference Acceleration in Diffusion Models.** Improving the efficiency of diffusion inference has become a growing area of focus. Sampling-based methods reduce denoising steps through distillation (Salimans & Ho, 2022), or step-aware training (Xiao et al., 2024). Architectural methods employ memory-aware attention, such as in HiDiffusion (Zhang et al., 2024a) and DiffIR (Xia et al., 2023). Model pruning and lightweight design strategies have also been proposed, as seen in Snap-Fusion (Li et al., 2023), Effortless Efficiency (Zhang et al., 2024b), and DiP-GO (Zhu et al., 2024). Patch-wise generation and conditional masking (Avrahami et al., 2023) have also been explored to control computation scope. However, most existing methods require retraining, loss rebalancing, or internal modifications to the model. Many of these approaches are orthogonal to our work and can be integrated in principle, but they do not account for the structural characteristics of sinogram data, such as directional sparsity and local interpretability. Applying them effectively in this domain would require additional adaptation.

**Memory and Runtime Efficient Deep Learning.** Beyond diffusion, memory- and latency-aware inference strategies have been widely studied in deep learning. Techniques include recomputation and memory reuse (Chen et al., 2016; Jain et al., 2020), and runtime-adaptive depth scaling (Xia et al., 2021). MEST (Yuan et al., 2021) also explore sparsity-aware training or deployment under tight memory budgets. These techniques address memory bottlenecks in standard feedforward networks and our work focuses instead on iterative generative models with different constraints.

## 3 HiSin: High-Resolution Sinogram Inpainting

As shown in Fig 1, HiSin builds upon RePaint (Lugmayr et al., 2022) as its default backbone and restructures the inference process into a three-stage resolution-guided pipeline. The sinogram is first denoised at low resolution to establish global structure, then refined at mid resolution, and finally completed at full resolution through patch-wise inference. At each stage, the upsampled output from the previous resolution is fused with the current input before denoising, ensuring hierarchical guidance across scales. This hierarchical design avoids full-frame activation and substantially reduces memory usage while preserving long-range consistency. To further improve efficiency under the structural characteristics of sinograms, HiSin introduces two inference-time modules: (1) frequency-aware patch skipping, which exploits the spectral sparsity of background regions to bypass redundant computation, and (2) structure-adaptive step allocation, which leverages local structural heterogeneity to adjust denoising depth per patch.

### 3.1 Resolution-Guided Progressive Inference

To efficiently inpaint high-resolution sinograms while avoiding memory overflow, HiSin adopts a three-stage progressive inference pipeline operating over low, mid, and high resolutions. Let $x_r$ denote the input at resolution $r \in \{low, mid, high\}$. Each resolution level $x_r$ is obtained by downsampling the original sinogram using a fixed ratio: low and mid-resolution inputs correspond to $0.25\times$ and $0.5\times$ the original resolution, respectively. At the first stage, full DDIM inference is performed on $x_{low}$ to generate a coarse prior $\hat{x}_{low}$, which captures global structural cues at minimal memory cost.

In the second stage, we refine the geometry at mid-resolution. $x_{mid}$ is fused with the upsampled output from the previous stage using a resolution-aware weighted sum:

$$\tilde{x}_{mid} = \lambda_{mid} \cdot Up(\hat{x}_{low}) + (1 - \lambda_{mid}) \cdot x_{mid}, \tag{1}$$

where $\lambda_{low} \in [0, 1]$ is a scalar controlling the fusion strength, and $Up(\cdot)$ denotes nearest-neighbor upsampling. This fusion helps the model retain mid-level structure while reinforcing global context. The resulting $\tilde{x}_{mid}$ is then denoised via DDIM to produce $\hat{x}_{mid}$.

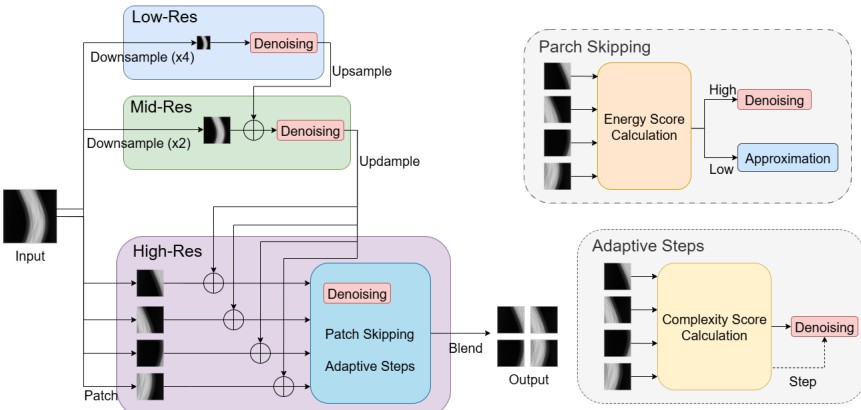

Figure 1: Overview of HiSin. The left illustrates the three-stage resolution-guided pipeline: low resolution followed by mid and high resolutions in a progressive refinement scheme, with the final stage performed patch-wise for detail recovery. At each stage, the upsampled output from the previous resolution is fused with the current input before denoising. The right zooms into the high-resolution stage, where two modules are applied: frequency-aware patch skipping (bypassing low-information patches) and structure-adaptive step allocation (assigning variable denoising steps by patch complexity).

At the final stage, we operate on the original full-resolution input. We divide $x_{high}$ into patches $\{x^i_{high}\}$. Each patch is of fixed size, ensuring consistent inference cost per patch across resolutions. For each patch, we retrieve the aligned region from $\hat{x}_{mid}$, upsample it, and fuse it with the local patch:

$$\tilde{x}^i_{high} = \lambda_{high} \cdot Up(\hat{x}^i_{mid}) + (1 - \lambda_{high}) \cdot x^i_{high}. \tag{2}$$

Each $\tilde{x}^i_{high}$ is then denoised via DDIM. Sec 3.2 and Sec 3.3 introduce more details about the final stage. This patch-wise processing avoids full-frame memory load while preserving long-range consistency through hierarchical conditioning. Each patch is processed sequentially during inference to ensure bounded memory usage.

## 3.2 FREQUENCY-AWARE PATCH SKIPPING

High-resolution sinograms often contain broad low-frequency backgrounds with narrow high-frequency details. To reduce computation in low-complexity regions, HiSin introduces frequency-aware patch skipping based on localized spectral content.

### 3.2.1 FREQUENCY-AWARE PATCH PRUNING

To reduce unnecessary computation in low-information regions of high-resolution sinograms, we introduce a mask-aware frequency-based patch skipping strategy. Let $P$ denote a high-resolution image patch, and let $\mathcal{F}(P)$ represent its real-valued 2D Fourier transform. We first compute the high-frequency energy ratio $\gamma(P)$ as:

$$\gamma(P) = \frac{\sum_{(u,v)\in\Omega_{high}} |\mathcal{F}(P)_{u,v}|^2}{\sum_{(u,v)} |\mathcal{F}(P)_{u,v}|^2}, \tag{3}$$

where $\Omega_{high}$ is a predefined high-pass band, typically the outer third of the frequency spectrum. In addition, we measure the mask ratio $r$. To incorporate mask awareness, we define the adjusted score as:

$$\gamma'(P) = (1 - r(P)) \cdot \gamma(P) + \tau \cdot r(P), \tag{4}$$

where $\tau$ is a small constant that safeguards against skipping heavily masked patches. A patch is considered redundant and skipped if its adjusted score $\gamma'(P)$ falls below a threshold $\tau$. In this way, structurally smooth or empty regions can still be skipped, while patches with large missing areas are preserved for reliable completion.

Rather than running full DDIM inference for spectrally sparse patches, we replace their outputs with a fixed approximation. It is obtained by passing a synthetic input patch–filled with low-amplitude Gaussian noise with mean 0, and standard deviation 0.01–through the original RePaint and DDIM steps. This design simulates typical background regions. It avoids repeated computation in trivial regions and ensures consistency with standard outputs.

### 3.2.2 COSINE-BASED PATCH BLENDING

To avoid visual artifacts at patch boundaries, particularly when adjacent regions are inferred with different mechanisms, we apply a smooth blending operation between neighboring patches. This is especially important in high-resolution sinograms where directional continuity must be preserved.

Let $P_1$ and $P_2$ denote two horizontally adjacent reconstructed patches, and $p \in [0, L]$ be local coordinate across boundary region of width $L$ pixels. Define a cosine-based spatial weight function:

$$\alpha(p) = \frac{1}{2} \left(1 - cos(\frac{\pi p}{L})\right), \tag{5}$$

which increases smoothly from 0 to 1 across the boundary. The blended pixel value is computed as:

$$P_{blend}(x, y) = \alpha(p) \cdot P_1(x, y) + (1 - \alpha(p)) \cdot P_2(x, y). \tag{6}$$

This produces a smooth interpolation where the transition is soft and visually imperceptible, reducing abrupt intensity jumps at patch edges.

We apply this blending only when necessary. Specifically, we compute the average gradient magnitude in the overlapping region using a standard Sobel filter applied to the corresponding mid-resolution sinogram. If the gradient exceeds a predefined threshold $\eta$, blending is enabled; otherwise, a simple hard stitch is used to save computation in flat areas. This conditional mechanism ensures efficiency without compromising visual consistency in structurally complex regions.

### 3.3 STRUCTURE-ADAPTIVE DENOISING

To further reduce computation, we allocate a different number of denoising steps to each patch based on its structural complexity. Unlike static diffusion sampling–where every region receives the same number of DDIM steps–we adapt the inference depth dynamically to match the content difficulty.

### 3.3.1 COMPLEXITY SCORE COMPUTATION

Let $P$ denote a patch in the high-resolution sinogram. We define a complexity score $\kappa(P)$ that combines two frequency-domain measures: Shannon entropy and high-frequency energy. The Shannon entropy $\mathcal{H}(P)$ is computed over the pixel intensity histogram:

$$\mathcal{H}(P_i) = -\sum_j p_j \log p_j, \tag{7}$$

where $p_j$ is the normalized count of pixel intensity in bin $i$. This captures texture randomness and distribution uniformity.

Also, we compute the spectral energy using the L1-norm of the 2D FFT of the patch. The overall complexity score is defined as:

$$\kappa_i = \mathcal{H}(P_i) + \log(1 + ||\mathcal{F}(P_i)||_1), \tag{8}$$

where $\mathcal{F}(P_i)$ is the 2D real-valued FFT of the patch and $||\cdot||_1$ denotes the L1-norm of the spectrum.

Entropy captures randomness and texture variation in the spatial domain, while the FFT energy captures signal richness and directional complexity. Their combination provides a robust, domain-agnostic proxy for structural difficulty.

### 3.3.2 PATCH-WISE STEP MAPPING

To convert $\kappa_i$ into a patch-specific number of denoising steps, we apply a sigmoid-based scaling function:

$$S_i = \lfloor S_{min} + (S_{max} - S_{min}) \cdot \sigma(\beta(\kappa_i - \mu)) \rceil \tag{9}$$

Table 1: Peak GPU memory (GB) and inference time (s) of different methods on `TomoBank` with random masks (ratio = 0.8) at resolutions 2048×2048 and 1024×1024. "OOM" indicates out-of-memory. Results on `LoDoPaB` exhibit nearly identical patterns in both memory and runtime and are omitted here for brevity.

| Method | 2048×2048 | | 1024×1024 | |
|---|---|---|---|---|
| | Peak Mem | Inf Time | Peak Mem | Inf Time |
| RePaint | OOM | | 18.7 | 2.20 |
| DiffIR | OOM | | 22.0 | 2.54 |
| HiDiffusion | 35.7 | 2.73 | 12.5 | 0.92 |
| HiSin (ours) | **24.7** | **2.25** | **8.7** | **0.75** |
| SinoTx | OOM | | 19.5 | 0.38 |
| ICT | OOM | | 17.0 | 0.20 |
| Spline (CPU) | 0.4 | 0.22 | 0.4 | 0.07 |
| Bilinear (CPU) | 0.3 | 0.12 | 0.3 | 0.04 |

where $\mu$ is the mean complexity across all patches in the sinogram; $\beta$ controls the steepness of the sigmoid transition; $\sigma(\cdot)$ is the standard sigmoid function; $S_{min}$, $S_{max}$ define the step range; $\lfloor \cdot \rfloor$ denotes rounding down to the nearest integer.

This ensures a soft but data-driven allocation: patches with below-average complexity receive fewer sampling steps, while others are modeled more deeply. Using the sigmoid ensures that step variation remains smooth, differentiable, and robust to outliers. The centering around the mean further normalizes the step distribution across sinograms with varying overall complexity.

## 4 EVALUATION

The evaluation has two main objectives: (1) demonstrating that HiSin significantly improves memory efficiency and inference speed while maintaining inpainting quality (Sec 4.2); (2) conducting ablation studies to validate the effectiveness of our mechanisms, including multi-resolution progressive inference, frequency-aware patch skipping and structure-adaptive denoising (Sec 4.3).

### 4.1 EXPERIMENTAL SETUP

**Evaluation Platform and Settings.** All experiments are performed on a single NVIDIA A100 GPU with 40 GB on-device memory from the Polaris supercomputer of Argonne Leadership Computing Facility (ALCF) resources at Argonne National Laboratory (ANL)[1], using CUDA 12.2, and PyTorch 2.1.0. All implementations employ PyTorch built-in optimizations: `torch.compile()` to enable graph-mode execution, mixed-precision inference via `torch.autocast()` (fp16), and `cudnn.benchmark = True` for kernel tuning.

All diffusion-based experiments use the denoising diffusion implicit models (DDIM) (Song et al., 2020) sampler with 50 steps. This step count is consistent with values evaluated in DDIM, where 50-step sampling achieves a strong balance between quality and efficiency. All inferences are performed with a batch size of 1. For high-resolution inference, we partition sinograms into overlapping patches with patch size $P = 128$, overlap $O = 32$, and stride $S = 96$. This setting balances efficiency and contextual coverage, while overlap mitigates boundary artifacts. For patch skipping, the high-frequency threshold is set to $\tau = 0.08$, which empirically filters low-information patches without affecting inpainting fidelity. For adaptive step scheduling, we use $(S_{min}, S_{max}) = (10, 50)$ to stay within the DDIM step budget while allowing fine-grained inference depth. More details about hyperparameters are in Appendix.

To ensure fair comparison, all baseline models are retrained from scratch using their official code-bases. Since HiSin operates entirely at inference time, no retraining or task-specific fine-tuning is applied. Each dataset provides 100,000 training samples at 512×512 resolution and 50 test samples.

---

[1]Polaris Cluster at ALCF: https://www.alcf.anl.gov/polaris

Table 2: SSIM / PSNR on `TomoBank` and `LoDoPaB` datasets under random and periodic masks (ratio = 0.6, 0.8) at resolutions 2048×2048 and 1024×1024. Each entry is formatted as: inpainted sinogram (CT reconstruction by FBP (Ramachandran & Lakshminarayanan, 1971)).

| Method | Mask | TomoBank | | Lodopab | |
|---|---|---|---|---|---|
| | | Ratio = 0.6 | Ratio = 0.8 | Ratio = 0.6 | Ratio = 0.8 |
| | | 2048×2048 | | | |
| HiSin | Random | **0.930** (0.916) / **30.9** (29.9) | **0.927** (0.913) / **30.6** (29.7) | **0.940** (0.926) / **31.6** (30.4) | **0.935** (0.924) / **31.3** (30.4) |
| | Periodic | **0.926** (0.913) / **30.6** (29.7) | **0.924** (0.910) / **30.3** (29.4) | **0.937** (0.922) / **31.3** (30.4) | **0.934** (0.924) / **31.0** (30.1) |
| HiDiffusion | Random | 0.903 (0.889) / 29.1 (28.2) | 0.900 (0.886) / 28.8 (27.9) | 0.911 (0.899) / 29.8 (28.9) | 0.910 (0.893) / 29.5 (28.6) |
| | Periodic | 0.902 (0.887) / 28.9 (28.0) | 0.897 (0.883) / 28.6 (27.7) | 0.912 (0.898) / 29.6 (28.7) | 0.904 (0.892) / 29.1 (28.3) |
| Spline | Random | 0.702 (0.688) / 23.8 (22.9) | 0.698 (0.684) / 23.5 (22.6) | 0.714 (0.694) / 24.5 (23.6) | 0.711 (0.690) / 24.3 (23.4) |
| | Periodic | 0.698 (0.681) / 23.7 (22.6) | 0.694 (0.680) / 23.2 (22.3) | 0.709 (0.694) / 24.4 (23.5) | 0.703 (0.690) / 23.9 (23.1) |
| Bilinear | Random | 0.695 (0.681) / 22.5 (21.7) | 0.691 (0.677) / 22.3 (21.4) | 0.703 (0.695) / 23.3 (22.6) | 0.703 (0.687) / 22.9 (22.2) |
| | Periodic | 0.695 (0.677) / 22.1 (21.4) | 0.687 (0.673) / 21.9 (21.1) | 0.703 (0.686) / 22.8 (22.0) | 0.695 (0.681) / 22.3 (21.5) |
| | | 1024×1024 | | | |
| HiSin | Random | **0.932** (0.919) / 31.2 (30.4) | 0.927 (0.911) / **30.9** (30.0) | **0.944** (0.929) / **32.1** (31.3) | **0.936** (0.924) / 31.1 (30.5) |
| | Periodic | **0.929** (0.916) / **30.9** (30.1) | 0.924 (0.913) / **30.6** (29.8) | **0.937** (0.924) / **31.5** (30.8) | **0.935** (0.922) / **30.9** (30.2) |
| RePaint | Random | **0.932** (0.919) / **31.4** (30.6) | 0.928 (0.915) / **30.9** (30.2) | **0.944** (0.927) / **32.1** (31.4) | **0.936** (0.925) / **31.2** (30.4) |
| | Periodic | **0.929** (0.914) / **31.1** (30.3) | **0.925** (0.912) / **30.8** (30.0) | **0.938** (0.924) / **31.8** (31.1) | **0.934** (0.924) / **31.0** (30.5) |
| HiDiffusion | Random | 0.902 (0.888) / 29.8 (29.0) | 0.898 (0.884) / 29.4 (28.6) | 0.914 (0.900) / 30.7 (29.8) | 0.909 (0.896) / 30.3 (29.5) |
| | Periodic | 0.899 (0.885) / 29.5 (28.7) | 0.895 (0.881) / 29.1 (28.3) | 0.910 (0.896) / 30.4 (29.6) | 0.906 (0.892) / 29.9 (29.1) |
| DiffIR | Random | 0.893 (0.881) / 29.4 (28.6) | 0.889 (0.877) / 29.0 (28.2) | 0.905 (0.892) / 30.2 (29.4) | 0.900 (0.888) / 29.8 (29.0) |
| | Periodic | 0.890 (0.878) / 29.1 (28.3) | 0.886 (0.874) / 28.7 (27.9) | 0.901 (0.889) / 29.7 (28.9) | 0.897 (0.885) / 29.3 (28.5) |
| SinoTx | Random | 0.880 (0.866) / 28.8 (28.0) | 0.876 (0.862) / 28.4 (27.6) | 0.891 (0.877) / 29.6 (28.7) | 0.887 (0.873) / 29.1 (28.2) |
| | Periodic | 0.877 (0.863) / 28.5 (27.7) | 0.873 (0.859) / 28.1 (27.3) | 0.888 (0.873) / 29.2 (28.4) | 0.884 (0.870) / 28.7 (27.9) |
| ICT | Random | 0.872 (0.858) / 28.5 (27.6) | 0.868 (0.854) / 28.0 (27.2) | 0.882 (0.869) / 29.0 (28.2) | 0.876 (0.864) / 28.6 (27.7) |
| | Periodic | 0.869 (0.855) / 28.1 (27.3) | 0.865 (0.851) / 27.7 (26.9) | 0.880 (0.866) / 28.7 (27.9) | 0.873 (0.862) / 28.3 (27.4) |
| Spline | Random | 0.718 (0.704) / 24.7 (23.8) | 0.713 (0.699) / 24.3 (23.4) | 0.728 (0.714) / 25.3 (24.4) | 0.723 (0.710) / 24.9 (24.2) |
| | Periodic | 0.715 (0.701) / 24.4 (23.5) | 0.710 (0.696) / 24.0 (23.1) | 0.726 (0.711) / 25.0 (24.1) | 0.721 (0.707) / 24.5 (23.7) |
| Bilinear | Random | 0.711 (0.697) / 23.4 (22.6) | 0.707 (0.693) / 23.0 (22.2) | 0.722 (0.708) / 24.1 (23.2) | 0.718 (0.702) / 23.7 (22.7) |
| | Periodic | 0.708 (0.694) / 23.1 (22.3) | 0.704 (0.690) / 22.7 (21.9) | 0.718 (0.704) / 23.7 (22.8) | 0.714 (0.700) / 23.2 (22.4) |

**Dataset.** We evaluate two real-world datasets `TomoBank` and `LoDoPaB`. `TomoBank` (De Carlo et al., 2018) consists of real sinogram images obtained from various materials and objects, derived from actual synchrotron radiation CT experiments, including Advanced Photon Source (APS)[2] at ANL. It also includes samples with dynamic features (Mohan et al., 2015) and in situ (Pelt & Batenburg, 2013) measurements, which capture a wide range of experiments at APS and other synchrotron facilities. To ensure consistency, we follow the official TomoPy (Gürsoy et al., 2014) preprocessing pipeline, which includes ring artifact removal, rotation center alignment, and normalization of intensities to the [0,1] range. Projection counts, angular sampling, and rotation centers are configured according to the metadata provided in TomoBank. `LoDoPaB` (Leuschner et al., 2021) is derived from the publicly available LIDC-IDRI lung CT database (Armato III et al., 2011). It provides simulated sinograms generated with a fixed parallel-beam geometry (1,000 projection angles, 513 detector bins), paired with real patient CT images. Unlike TomoBank, where sinograms are physically measured, LoDoPaB contains numerically generated sinograms with official splits provided via the DIVAL benchmark library (Leuschner et al., 2025). To simulate realistic sparse-view CT scenarios, we apply two types of angular masks: (1) random masks, which discard a random subset of projection angles, and (2) periodic masks, which regularly subsample angles at fixed intervals. The mask ratio denotes the fraction of angles removed (e.g., 0.8 means only 20% of the projection angles are retained). In this work, we mainly evaluate challenging cases with mask ratios of 0.6 and 0.8, where the majority of angular information is missing.

**Metrics.** We evaluate each method using both efficiency and quality metrics. For computational performance, we report peak GPU memory usage and inference runtime, measured over multiple forward passes. Peak memory reflects the maximum GPU allocation during inference, indicating worst-case hardware demand. For inpainting fidelity, we use Structural Similarity Index (SSIM) (Wang et al., 2004) and Peak Signal-to-Noise Ratio (PSNR). All images are normalized to the [0, 1] range before metric computation. SSIM is computed in single-scale mode using a 11×11 Gaussian kernel ($\sigma = 1.5$). All metrics are computed on grayscale images. SSIM captures perceptual and structural consistency, while PSNR quantifies absolute pixel-wise differences.

---

[2]Advanced Photon Source: https://www.aps.anl.gov

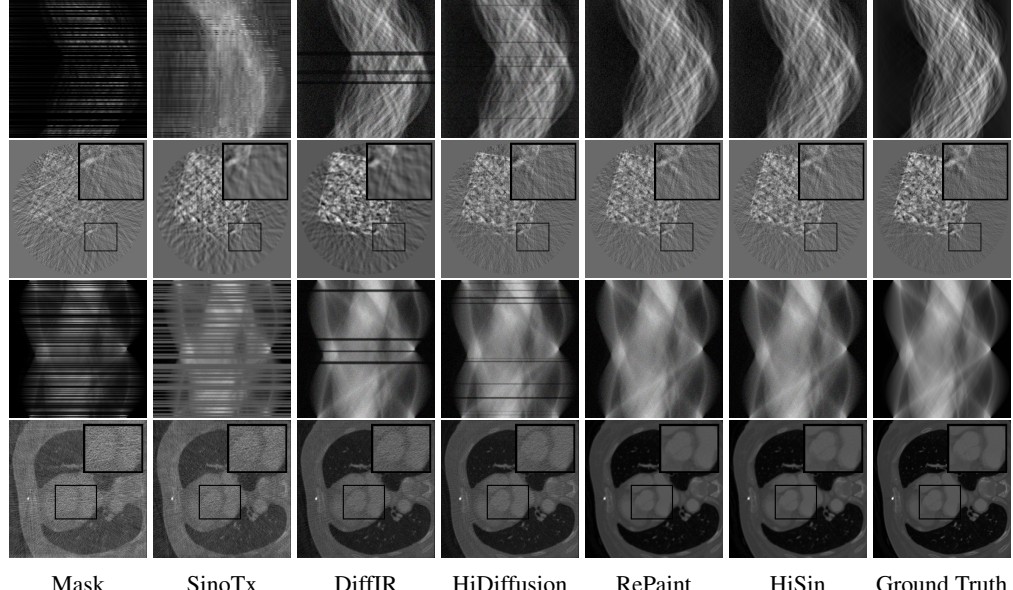

Figure 2: Qualitative inpainting results on `TomoBank` (lines 1 to 2) and `LoDoPaB` (lines 3 to 4) with random mask (ratio = 0.8) at 1024×1024 resolution. Odd columns and even columns show the sinograms and reconstructed images, respectively.

**Baselines.** We evaluate HiSin against a set of representative baselines, including diffusion-based models optimized for high-resolution inpainting, transformer-based methods, and classical interpolation techniques. Among diffusion models, we consider RePaint (Lugmayr et al., 2022), a general-purpose inpainting model that serves as the backbone for HiSin; HiDiffusion (Zhang et al., 2024a), which improves memory efficiency through resolution-aware U-Net design and localized attention; and DiffIR (Xia et al., 2023), which introduces a compact intermediate prior to reduce sampling steps for image restoration. For transformer-based approaches, we include SinoTx (Jiaze et al., 2025), tailored to sinogram completion, and ICT (Wan et al., 2021), designed for efficient inpainting. Finally, Bilinear and Spline interpolation serve as classical non-learning baselines.

## 4.2 OVERALL QUANTITATIVE AND QUALITATIVE RESULTS

**Peak Memory Usage Comparison.** As shown in Tab 1, HiSin significantly reduces peak GPU memory usage compared to prior methods across both tested resolutions. RePaint and DiffIR both encounter out-of-memory (OOM) failures at 2048×2048 resolution, indicating that standard diffusion pipelines cannot scale to such resolutions even with a high-end GPU. In contrast, HiSin remains fully operational under these conditions and achieves up to a 30.81% reduction in memory usage relative to the most efficient baseline HiDiffusion. Notably, this trend holds consistently across input sizes, indicating that our framework scales gracefully.

**Inference Time Comparison.** As shown in Tab 1, HiSin consistently achieves faster inference than the fastest diffusion-based baseline, HiDiffusion, with speed up to 17.58% at 2048×2048 resolution. These improvements stem from two sources: progressive scheduling reduces redundant steps in simple regions, while patch skipping eliminates computation entirely for spectrally sparse areas. We further validate the effectiveness of these two mechanisms in the ablation studies (Sec 4.3). This demonstrates that high-resolution efficiency can be achieved without sacrificing execution time.

**Inpainting Quality.** Tab 2 summarizes accuracy results. At 2048×2048 resolution, HiSin achieves the best performance among all baselines while remaining memory-efficient, demonstrating its ability to extend high-quality inpainting to resolutions where other diffusion models fail. At 1024×1024, HiSin delivers accuracy comparable to its computation-intensive counterpart RePaint, showing that our optimizations do not compromise fidelity at moderate scales. Compared to DiffIR and HiDiffusion, HiSin consistently achieves higher SSIM and PSNR across mask ratios, with

Table 3: Ablation study of the three key components: multi-resolution progressive inference, adaptive denoising, and patch skipping. Results are reported as peak GPU memory (GB), inference time (s), and SSIM/PSNR on `TomoBank` with random masks (ratio = 0.8) at resolution 2048×2048.

| Method | Peak mem | Inf time | SSIM | PSNR |
|---|---|---|---|---|
| HiSin | 24.7 | 2.25 | 0.927 | 30.6 |
| W/o low | 24.7 | 2.15 | 0.901 | 28.7 |
| W/o mid | 15.0 | 1.80 | 0.893 | 28.2 |
| W/o high | 24.5 | 1.20 | 0.870 | 27.1 |
| W/o adaptive steps | 24.7 | 3.05 | 0.928 | 30.8 |
| W/o patch skipping | 24.7 | 3.25 | 0.929 | 30.8 |

Table 4: Orthogonality study on `TomoBank` with random masks (ratio = 0.8) at resolution 2048×2048.

| Method | Peak mem | Inf time | SSIM | PSNR |
|---|---|---|---|---|
| HiDiffusion | 35.7 | 2.73 | 0.900 | 29.4 |
| HiDiffusion+HiSin | 20.0 | 1.85 | 0.900 | 29.2 |
| DiffIR | | OOM | | |
| DiffIR+HiSin | 24.5 | 2.0 | 0.883 | 28.6 |

improvements up to +0.03 SSIM and +1.8 dB PSNR. Fig 2 visualizes sinogram inpainting and reconstructed images, where HiSin produces nearly indistinguishable results from RePaint. These findings confirm that HiSin fundamentally extends diffusion-based inpainting to 2048×2048 resolution in a more memory- and runtime-efficient manner.

### 4.3 ABLATION STUDIES

To analyze the impact of individual components in HiSin, we perform ablation studies targeting its three core mechanisms: multi-resolution progressive inference, adaptive denoising based on structural complexity, and sparse patch skipping based on spectral sparsity. Tab 3 summarizes the results. Removing any resolution stage substantially degrades inpainting quality, indicating that different resolutions provide complementary benefits and that all stages are necessary to maintain overall fidelity. We note that peak memory is dominated by the mid-resolution stage, since high-resolution inference is performed patch-wise and patches are processed sequentially. Removing adaptive denoising or patch skipping does not affect SSIM/PSNR, but results in significant efficiency loss: inference time increases by 35.6% without adaptive steps (all set to 50) and by 44.4% without patch skipping, while memory usage remains unchanged. Together, these results validate the complementary nature of our three mechanisms: progressive inference balances global and local consistency, adaptive denoising reduces computational depth, and patch skipping reduces spatial workload, jointly enabling efficient high-resolution inpainting.

### 4.4 COMPATIBILITY WITH EXISTING OPTIMIZATIONS

Our optimizations are orthogonal to existing efficiency-oriented diffusion designs. HiSin is built upon RePaint as its default backbone, but our optimizations operate entirely at the inference stage and can be applied to other diffusion architectures. To demonstrate this, we combine HiSin with two recent baselines, HiDiffusion and DiffIR. As shown in Tab 4, integrating HiSin further reduces peak memory and inference time, while maintaining comparable inpainting quality.

## 5 CONCLUSION

We present HiSin, a novel framework for efficient high-resolution sinogram inpainting. HiSin integrates mechanisms that adaptively allocate computation across resolutions, spatial regions, and inference depth. This design enables inpainting on 2048×2048 sinograms using a single GPU, significantly reducing memory usage and runtime while maintaining the inpainting fidelity.

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
