# A    ADDITIONAL DESIGN NOTES ABOUT HYPERPARAMETER CHOICES

## A.1    SENSITIVITY OF THE PATCH-SKIPPING THRESHOLD

### A.1.1    RATIONALE BEHIND THE PATCH-SKIPPING THRESHOLD SELECTION

The parameter $\tau$ controls the patch-skipping threshold in HiSin. Specifically, it determines the spectral energy level below which a patch is considered uninformative and thus can be skipped during inference. A larger value of $\tau$ results in more aggressive skipping, while a smaller value leads to fewer patches being skipped and then higher inference time.

We set $\tau = 0.08$ in all our experiments based on three practical considerations:

- Empirical coverage of low-frequency components: In typical sinogram inputs, a threshold of 0.08 allows us to skip patches with minimal high-frequency activity while still retaining structurally important information.

- Interpretability and reproducibility: Compared to more aggressive thresholds, the 0.08 setting avoids dropping structurally ambiguous or borderline patches.

- Stability across datasets and resolutions: We observed that $\tau = 0.08$ yields stable performance across all datasets, and across multiple input resolutions and mask ratios. This suggests that the choice is not overly sensitive to dataset-, input resolution- or mask ratio-specific properties.

These empirical observations collectively support the choice of $\tau = 0.08$ as a robust and balanced default.

### A.1.2    SENSITIVITY ANALYSIS EXPERIMENTS

Table 5: Sensitivity of HiSin to different values of $\tau$ on `TomoBank` with random masks (ratio = 0.8) at resolution 2048×2048. SSIM and PSNR are formatted as: inpainted sinogram (reconstructed CT image by FBP).

| $\tau$ | SSIM | PSNR |
|------|-------------|-------------|
| 0.12 | 0.912 (0.908) | 29.7 (28.5) |
| 0.08 | 0.927 (0.913) | 30.6 (29.7) |
| 0.05 | 0.928 (0.913) | 30.6 (29.8) |

Tab 5 reports evaluation results for $\tau$ sensitivity. We compare SSIM and PSNR under three representative values of $\tau$: 0.12, 0.08, and 0.05. These thresholds respectively correspond to aggressive skipping, default setting, and conservative skipping.

We observed that decreasing $\tau$ beyond 0.08 yields diminishing returns in inpainting accuracy, while causing an increase in inference time due to the larger number of patches retained. Therefore, we set $\tau = 0.08$ as a practical balance point—-preserving computational efficiency without sacrificing output quality.

## A.2    CHOICES OF OTHER HYPERPARAMETERS

**Resolution fusion weights $\lambda_{mid}$ and $\lambda_{high}$.** Both $\lambda_{mid}$ and $\lambda_{high}$ are set to 0.5, balancing the influence of lower-resolution priors including global features with high-resolution structural details during progressive inference.

**Gradient threshold $\eta$ for patch blending.** We set $\eta = 0.1$ to balance efficiency and visual quality in patch stitching. A lower threshold would lead to unnecessary computation with negligible benefit. A higher threshold risks skipping blending in structurally significant areas, causing visible seams.

**Sigmoid sharpness factor $\beta$.** We set the sigmoid sharpness factor $\beta = 5$ to achieve a smooth but sufficiently discriminative mapping from complexity scores to denoising step counts.

## A.3 DISTINCTION BETWEEN PATCH SKIPPING AND ADAPTIVE STEP ALLOCATION

While both patch skipping and adaptive step allocation rely on local complexity analysis, they serve distinct purposes and operate under different assumptions. Patch skipping is a coarse-grained gating mechanism that identifies structurally trivial regions where full denoising can be omitted entirely. It uses a simple frequency-domain criterion—-the ratio of high-frequency energy—-to detect spectrally sparse patches dominated by very smooth or empty content. This decision is binary and conservative: a skipped patch will never undergo any denoising, so the threshold must be set cautiously to avoid quality degradation.

In contrast, adaptive step allocation aims to adjust the computational effort spent on patches that are retained for inference. Instead of deciding whether to process a patch, it modulates how many DDIM steps to apply based on a richer complexity score. This score combines Shannon entropy over pixel intensities with the total frequency energy (via the L1 norm of the FFT), capturing both spatial irregularity and spectral richness. The resulting step count is assigned through a smooth sigmoid mapping centered around the mean complexity across all patches, ensuring stable and data-aware inference depth scheduling.

While both mechanisms involve complexity estimation, their criteria, granularity, and functional roles are fundamentally different.

## B LIMITATIONS: COMPUTATIONAL COST ANALYSIS

Table 6: FLOPs (G) for different methods on `TomoBank` with random masks (ratio = 0.8) at resolutions 2048×2048 and 1024×1024. All FLOPs are estimated via code-level calculation based on resolution, patch size, and denoising steps. These calculations are performed using standard layerwise formulas, without reliance on hardware profiling.

| Method | 2048×2048 | 1024×1024 |
|---|---|---|
| RePaint | OOM | 34.7 |
| DiffIR | | 22.5 |
| HiDiffusion | 112.6 | 28.4 |
| HiSin | 118.3 | 29.6 |

While HiSin improves memory usage and inference speed through hierarchical scheduling and patch-level optimization, it comes with a moderate increase in total FLOPs compared to other efficiency-oriented baselines. As shown in Tab 6, HiSin incurs more FLOPs than HiDiffusion and DiffIR. This reflects the structural overhead of progressive inference and adaptive patch processing. While the FLOPs slightly increase compared to HiDiffusion and DiffIR, this trade-off is necessary to support higher resolution sinogram inpainting, while maintaining high memory efficiency and inpainting quality.