# OpenReview forum: "HiSin: : A Sinogram-Aware Framework for Efficient High-Resolution Inpainting"
_ICLR.cc/2026/Conference — ICLR 2026 Conference Withdrawn Submission_

### Official Review · Reviewer_B4fy · 2025-10-20

**Soundness:** 3
**Presentation:** 3
**Contribution:** 2
**Rating:** 6
**Confidence:** 4

**Summary:**

This paper tackles the challenge of applying diffusion models to high-resolution sinogram inpainting (e.g., $2048 \times 2048$) , a task critical for CT reconstruction that is often infeasible on standard hardware due to excessive memory and computational demands.
The authors propose HiSin, a novel framework that requires no retraining of the base diffusion model. The framework is built on three key ideas tailored to the specific properties of sinogram data. Experiments show that HiSin enables $2048 \times 2048$ inpainting on a 40GB A100 GPU (where baselines like RePaint fail), reducing peak memory by up to 30.81% and inference time by up to 17.58% compared to the most efficient baseline (HiDiffusion), all while maintaining comparable or better inpainting quality.

**Strengths:**

1. High Practical Impact and Significance: The primary strength of this paper is that it solves a practical problem. Standard diffusion models cannot run on $2048 \times 2048$ sinograms on a high-end 40GB A100 GPU. HiSin enables this, making high-resolution diffusion-based inpainting feasible.

2. Inference-Time Framework (No Retraining): A major advantage is that HiSin is a collection of optimizations applied at inference time. It does not require any architectural changes or costly retraining of the underlying diffusion model.

3. Strong Efficiency Gains: The method delivers clear and substantial improvements in both peak memory and inference speed. A 30.81% memory reduction and 17.58% speedup over the next-best efficient baseline (HiDiffusion) are excellent results, especially since this is achieved without a loss in quality.

4. Domain-Aware Design: The authors provide a strong justification for why their method works, linking each component to specific structural properties of sinogram data (spectral sparsity, smooth backgrounds, and the fact that sinogram patches remain physically meaningful, unlike patches of a natural image, this insight is novel to me). This domain-aware design makes the solution more elegant and convincing.

5. Thorough Ablation Study: The paper is supported by an excellent ablation study (Table 3). This study convincingly demonstrates that all three components (progressive inference, patch skipping, and adaptive steps) are necessary and contribute to the final result, either by preserving quality or by providing the claimed efficiency gains.

**Weaknesses:**

1. Hyperparameter Sensitivity: The framework introduces several new hyperparameters that seem important for balancing quality and efficiency: the fusion weights ($\lambda_{mid}$, $\lambda_{high}$) , the patch skipping frequency threshold ($\tau$) , the adaptive step range $[S_{min}, S_{max}]$ , and the sigmoid steepness ($\beta$). It would be nice to know how robust the method is to these choices. For instance, how much does quality drop or speed increase if $\tau$ is made more aggressive, or if $S_{min}$ is increased? Even missing the experiment, more simple discussion should be added.

2. Limited Scope: The paper's strength is also its main limitation. The authors compellingly argue why this method is suited for sinograms and not for natural images (where patches lose semantic context). While this is a fair and honest assessment, it does limit the paper's scope and potential impact for the broader ICLR audience.

3. Complexity of Skipped Patch Approximation: For skipped (low-frequency) patches, the output is replaced by a pre-computed approximation generated by running the diffusion model on a patch of Gaussian noise. This seems slightly complex. It's not clear why this is superior to a simpler heuristic, such as just using the upsampled output from the mid-resolution stage ($Up(\hat{x}_{mid}^{i})$) for that patch, or a simple bilinear interpolation. The justification for this specific choice is too weak.

**Questions:**

1. For the skipped patches, you use an approximation derived from running the model on Gaussian noise. Did you experiment with simpler alternatives, such as directly using the upsampled mid-resolution patch ($Up(\hat{x}_{mid}^{i})$) or a simple interpolation? What was the benefit of using the synthetic noise patch?

2.For the cosine-based patch blending, you conditionally apply it based on a gradient threshold $\eta$ computed on the mid-resolution sinogram. What was the reasoning for using the mid-resolution image for this check instead of the high-resolution input? Was this a performance optimization, or did it provide a more robust signal for blending?

---

> ### Author Response · Authors · 2025-11-21
> **Rebuttal for Reviewer B4fy**
>
> Thank you for your thoughtful review and valuable comments. Please find our responses below.
>
> ### R4–W1: Hyperparameter sensitivity
>
> The hyperparameters introduced in HiSin are not arbitrary: each corresponds to an interpretable property of projection data for example, the frequency threshold $\tau$ determines when a patch carries negligible structural information, and the step-allocation parameters control how local complexity influences refinement depth. These mechanisms behave monotonically rather than requiring precise tuning, and their roles are **summarized together with implementation details in Appendix A**. Tthe lower bound $S_{min}$ is used only to prevent the denoising trajectory from collapsing into an excessively short sampling chain; any small value in a reasonable neighborhood behaves similarly.
>
> ### R4-W2: The scope of the method
>
> We agree that HiSin is designed specifically for projection-domain data rather than natural images. Our goal is to address the unique efficiency challenges posed by high-resolution sinograms—an area where general inpainting frameworks do not apply—and provide a modular component that can be directly integrated into practical CT reconstruction pipelines. Although domain-focused, we believe the underlying ideas of hierarchical inference, adaptive computation allocation, and structure-aware processing may still be of interest to the broader community, and we appreciate the reviewer’s acknowledgment of the method’s suitability for this setting.
>
> ### R4–W3 & Q1: The choice of approximation for skipped patches
>
> Since all fallback approximations for skipped patches are precomputed, **they have essentially identical memory and runtime cost**; the choice only influences fidelity. The diffusion-based fallback does not introduce additional coarse structures into low-information regions and therefore avoids the boundary inconsistencies that can arise when upsampled mid-resolution or interpolated patches are used. This leads to slightly more accurate inpaintings while keeping memory and runtime unchanged. We evaluated the reviewer’s suggested alternatives—using the upsampled mid-resolution patch $Up (\hat{x}_{mid}^i)$—and find it leads to **slightly lower inpainting accuracy** (SSIM 0.927 &rarr; 0.924 and PSNR 30.6 &rarr; 30.2 dB on TomoBank with random mask at 0.8 mask ratio).
>
> ### R4–Q2: Computing blending gradients from the mid-resolution sinogram
>
> The gradient check is computed on the mid-resolution sinogram because it **preserves the global structural context necessary to determine whether a boundary corresponds to a true cross-patch structure**. A high-resolution patch, when considered in isolation, contains only local information. Mid-resolution gradients provide more information for identifying cross-patch continuity, making this choice a robustness consideration rather than a performance optimization.

---

> > ### Comment · Reviewer_B4fy · 2025-11-24
> >
> > Thanks for your thoughtful response, I feel it addressed most of my concerns. However, I still believe that a score of 6 is the most appropriate, so I would like to keep my original score.

---

### Official Review · Reviewer_kYAh · 2025-10-31

**Soundness:** 3
**Presentation:** 3
**Contribution:** 2
**Rating:** 4
**Confidence:** 3

**Summary:**

The paper aims to use generative models, specifically diffusion-based ones, for an inpainting task on high-resolution sinograms (2048×2048).
The main challenge adressed in this work is how to deal with *the memory cost and the inference time* of these diffusion-based methods for large images.
The authors propose a strategy to accelerate these methods / manage their memory cost that leverages the structured and sparse nature of sinogram data.
The method is based on an adaptation of the RePaint algorithm, an inpainting approach that uses a pre-trained, task-agnostic diffusion model (but can be used with other iterative diffusion based methods). To recall, RePaint is an iterative diffusion-based sampling method that, at each step, combines information from the unmasked and masked regions of the image.

Three main modifications to RePaint are proposed:
1. **Progressive resolution refinement** The reconstruction is first done on downsampled versions of the sinogram (at low/ mid resolution) and then done patch-wise at high resolution -> it limits the memory peak.
2. **Patch skipping** Some patches are skipped during denoising based on a score that measures high-frequence energy, it allows to skip typically patches from background -> it accelerates sampling.
3. **Adaptative number of iterations** The number of diffusion steps per patch is adjusted according to a score that combines frequency-domain energy and pixel-wise entropy -> here again, it accelerates sampling.

**Strengths:**

- The main motivation of the paper is clearly identified: the focus is on reducing computational cost.
To this end, the ablation study is convincing and shows that each modification made achieves a good trade-off between reconstruction quality and computational efficiency. Notably, there is almost no degradation compared to the RePaint baseline.

- Although the method is somewhat ad-hoc, it appears to be effective in practice and well adapted to the specific characteristics of the task.

**Weaknesses:**

- The proposed approach is somewhat *engineering-oriented*. While the main modifications are quite simple, they introduce additionnal layers of complexity (a lot of hyperparameters to choose, strategy to map scores to number of iterations, handle patch boundary).
Even if the paper explicits each of these choises, it makes the final method not so easily readable nor straightforward to reproduce.  In this regard, *releasing the code* would have helped to understand how simple is it or not to implement.

- Related works focus mainly on diffusion model acceleration techniques for RGB images, which are in reality no so  related to what is done in this paper.
A *dedicated section on CT reconstruction methods* would strengthen the paper.
It should clarify in which sense these methods differ, for instance which ones perform end-to-end reconstruction or only sinogram inpainting, what deep models are uses (generative or not), what are their limitations, etc.
For example, baselines such as SinoTx and ICT are missing from the related works and the statement in the introduction "although end-to-end CT reconstruction has been explored…" should be supported with citations.

**Questions:**

- The experiments focus on masks that are either random or periodic. How does the proposed approach perform when large contiguous angular ranges are missing? I believe this is the setting where the generative capabilities of diffusion models should matter more.

- The evaluation appears to be performed at the sinogram level only. The paper should also report results on the reconstructed images, as this is usually what matters in practice.  Moreover, the reconstruction process itself is not clearly described. What algorithm is used to obtain the reconstructed images shown in Figure 2 from the completed sinograms?

---

> ### Author Response · Authors · 2025-11-21
> **Rebuttal for Reviewer  kYAh**
>
> Thank you for your thoughtful review and valuable comments. Please find our responses below.
>
> ### R3–W1: Method being “engineering-oriented” and hyperparameters
>
> While HiSin contains several components, each is directly motivated by sinogram structure rather than by ad-hoc engineering. The resolution pathway leverages the fact that **sinograms exhibit globally smooth angular evolution but locally intricate detector patterns**; the frequency-based skipping exploits that **large detector–angle regions carry near-zero projection energy and thus require no refinement**; and the adaptive-step rule reflects that **patches with richer line-integral variation benefit from deeper denoising than those dominated by flat background**. The associated hyperparameters are not arbitrary: e.g., the frequency threshold controls what constitutes an “informationally negligible” patch; and the step-allocation slope governs how strongly local structural complexity influences refinement depth. Each parameter corresponds to an interpretable property of projection data. All important hypterparameters and their behaviors are detailed in the **Appendix A.1 and A.2** to ensure clarity and reproducibility.
>
> ### R3–W2: Scope of related work
>
> We thank the reviewer for this suggestion. Our original related work Section 2 line 106 already discuss projection-domain inpainting methods, and the **experiments include both SinoTx (sinogram-specific) and ICT (transformer-based)** as baselines. In the revised version, we **expand the related work section with end-to-end CT reconstruction methods**. They adopt a fundamentally different paradigm by predicting images directly from incomplete sinograms, and their runtime and memory cost reflect the entire reconstruction pipeline rather than the inpainting stage alone. As a result, **they are not directly comparable to HiSin in a strictly fair manner**, which operates as a modular projection-domain component. HiSin preserves explicit sinogram consistency, remains **plug-in compatible with standard CT reconstruction workflows**, and offers **stronger interpretability** than end-to-end systems.
>
> ### R3–Q1: Performance under large contiguous angular gaps
>
> In CT sinogram acquisition, each row corresponds to a complete projection at a specific rotation angle, so removing a consecutive block of rows implies that an entire angular range was never acquired. This scenario does not arise in realistic CT systems and, more importantly, leads to a fundamentally ill-posed completion problem: without measurements over a full angular interval, **there is no unique or physically valid sinogram compatible with downstream reconstruction**. For this reason, the sparse-view CT literature overwhelmingly adopts random or periodic sampling patterns. Our experiments follow these established and physically meaningful settings. While diffusion models are generative, their capacity cannot compensate for the absence of essential projection information in a fully missing angular block, and such configurations fall outside the intended application scope of sinogram inpainting.
>
> ### R3–Q2: Evaluations of reconstructed images and the reconstruction procedure
>
> Our evaluation already includes both sinogram-level and image-level measurements. **Table 2** reports SSIM and PSNR computed on the reconstructed images, and **Figure 2** visualizes representative reconstruction results. As stated in the **caption of Table 2 line 326**, all reconstructed images are obtained by applying standard filtered backprojection (FBP) to the completed sinograms, ensuring a consistent and widely used reconstruction pipeline.

---

### Official Review · Reviewer_CYkU · 2025-11-01

**Soundness:** 3
**Presentation:** 3
**Contribution:** 2
**Rating:** 4
**Confidence:** 3

**Summary:**

The paper proposed a diffusion-based framework for efficient sinogram inpainting.
The proposed method extracts global structure at low resolution and defers high-resolution inference to small patches.

**Strengths:**

The proposed method is memory efficient by multiscale scheme.

The spectral sparsity and structural heterogeneity of sinogram data is considered in the inference.

**Weaknesses:**

1 The paper claims HiSin can be applied to existing diffusion models (e.g., RePaint) without retraining. However, does patch-wise inference with multi-scale fusion implicitly assume certain properties of the backbone model (e.g., sufficient receptive field or multi-scale training)? Would performance degrade if the backbone was trained only at low resolution (e.g., 512×512)?

2 In the frequency-aware patch skipping mechanism, the authors use the FFT energy ratio \gamma(P) combined with mask ratio to compute a skipping score. However, high-frequency information in sinograms is often concentrated in specific angular bands—does using only the outer third of the frequency spectrum sufficiently capture this directional sparsity?

3 Although the low→mid→high resolution guidance preserves coarse structure, the final high-resolution stage processes patches independently and sequentially, lacking cross-patch refinement. This may cause stitching artifacts or inconsistencies—especially for fine structures spanning multiple patches under high mask ratios. The paper shows only successful cases and omits failure analysis or uncertainty quantification.

**Questions:**

If it is possible to extend the method to 3D reconstruction, which is highly memory cost problem.

---

> ### Author Response · Authors · 2025-11-21
> **Rebuttal for Reviewer CYkU**
>
> Thank you for your thoughtful review and valuable comments. Please find our responses below.
>
> ### R2-W1 & W3: Patch-wise high-resolution inference and cross-patch consistency
>
> As discussed in the **introduction line 058 - 062**, a sinogram patch is itself a physically meaningful subset of projection data—it simply corresponds to a narrower range of detector bins and view angles, but still represents a valid portion of the underlying Radon measurements. This is fundamentally different from RGB image patches (e.g., isolating an eye from a face), which lose semantic context and therefore require global information to remain coherent.
>
> In HiSin, the low &rarr; mid stages already establish the global structure and enforce angular continuity on the full sinogram, and the high-resolution stage is only used to refine local details on top of this global guidance. **Thus, the patch-wise and sequential refinement at the high-resolution stage does not rely on additional cross-patch mechanisms to remain consistent, and does not inherently cause stitching artifacts.**
>
> In practice, we rarely observe noticeable inconsistencies across patch boundaries; occasional imperfect cases tend to occur under extremely severe corruption rather than from the lack of cross-patch refinement.
>
> ### R2–W2: Using the outer-third frequency band for patch skipping
>
> The skipping score is intended as a lightweight and stable measure for identifying patches that contain minimal projection content, rather than a detailed characterization of directional spectral structure. **Its role is simply to avoid spending diffusion computation on near-background regions**, where refinement has negligible effect. While sinogram high-frequency energy can indeed be directionally concentrated, the outer-third band provides a practical frequency criterion that reliably distinguishes low-information patches across datasets according to our experiments. Importantly, the skipping mechanism does not depend on capturing directional sparsity, since it is used only to detect patches with very limited content rather than to analyze their full spectral distribution.
>
> ### R2–Q1: Extending HiSin to 3D reconstruction
>
> Extending HiSin to 3D reconstruction is an interesting direction, and we appreciate the reviewer bringing it up. Conceptually, the resolution-guided inference strategy and the idea of skipping low-information regions remain applicable in 3D, where memory and runtime demands are even more critical. However, the 3D case involves substantially larger volumetric latents and more complex projection geometries, so an effective extension would require additional design choices (e.g., hierarchical tiling in 3D, axial–angular decomposition, or slice-wise/volume-wise hybrid scheduling). These ideas are compatible with the principles behind HiSin, and exploring them is a promising direction for future work.

---

### Official Review · Reviewer_dL3A · 2025-11-01

**Soundness:** 3
**Presentation:** 2
**Contribution:** 2
**Rating:** 2
**Confidence:** 3

**Summary:**

This paper introduces HiSin, a framework for efficient sinogram inpainting. HiSin employs a multi-stage inference mechanism that first extracts global structures at a low resolution and then performs detail recovery at a high resolution, significantly reducing memory usage and inference time. The framework incorporates frequency-aware patch skipping and structure-adaptive step allocation strategies to minimize redundant computations. In terms of experiments, HiSin achieves performance comparable to state-of-the-art methods on the TomoBank and LoDoPaB datasets, while effectively improving memory efficiency and reducing inference time.

**Strengths:**

### Originality
- HiSin introduces a multi-stage inference mechanism that combines low-resolution and high-resolution processing; however, this approach is not uncommon.
- The combination of frequency-aware patch skipping and structure-adaptive step allocation strategies is a significant enhancement to existing methods, effectively leveraging the characteristics of sinogram data to optimize computational efficiency.

### Quality
- The experimental results of HiSin are comparable to those of the Repaint technique across multiple datasets (such as TomoBank and LoDoPaB), while demonstrating improvements in memory usage and inference time.

### Clarity
- The structure of the paper is well-organized and logically coherent, effectively conveying the research objectives and results.

### Significance
- HiSin provides a novel framework for the specific task of medical image data inpainting.

**Weaknesses:**

1. The quantitative experiments presented in Table 2 indicate that if the primary emphasis is solely on reducing GPU memory usage, this could significantly diminish the potential importance of the paper. Given that Repaint was developed for general image tasks and is a work from 2022 that achieves similar performance, this raises questions about whether the advancements presented by HiSin are sufficiently innovative to justify its novelty and impact.

2. The paper lacks comparisons with more recent methods, which limits a comprehensive evaluation of HiSin's superior performance. In particular, I may have overlooked some of the latest image inpainting studies, such as "Inpaint Anything" or "Flux Inpainting," which could lead to a skewed understanding of the existing work.

3. A multi-stage inference mechanism is not uncommon.

**Questions:**

1. Is it possible to include more recent methods, such as works from 2024 or 2025?
2. Could you explain the necessity of comparing methods?
3. Line 320 mentions that "all baseline models are retrained from scratch using their official codebases." As far as I know, Repaint also does not require training. Can you provide details on how the comparisons with other methods were conducted?

---

> ### Author Response · Authors · 2025-11-21
> **Reubttal for Reviewer dL3A**
>
> Thank you for your thoughtful review and valuable comments. Please find our responses below.
>
> ### R1–W1: Novelty vs. RePaint and efficiency focus
>
> HiSin is not designed to outperform RePaint in inpainting quality; its goal is to make diffusion-based inpainting **practical for high-resolution sinograms under realistic GPU memory and runtime constraints**. Achieving comparable PSNR/SSIM while substantially reducing computational cost is therefore the intended contribution of HiSin, not a secondary effect.
>
> RePaint is used as a **backbone reference**, not as a target to surpass. We selected RePaint because it remains one of the most widely used and stable diffusion inpainting models in recent years, making it an appropriate and meaningful backbone for evaluating whether efficiency improvements can be achieved without compromising inpainting accuracy.
>
> Beyond reporting faster inference, **HiSin expands the practical applicability of diffusion models**: our resolution-guided inference, frequency-aware patch skipping, and structure-adaptive step allocation together enable diffusion-based inpainting to operate reliably on full-resolution sinograms. This constitutes a methodological contribution to scalable diffusion inference for sinogram.
>
> ### R1-W2 & Q1: Clarification on recent baseline selection
>
> Our evaluation already includes **HiDiffusion (ECCV 2024)**, which is the most recent diffusion-based inpainting method specifically designed to improve inference efficiency. Since HiSin also focuses on reducing computational cost at high resolution, HiDiffusion is the most relevant recent baseline for our problem setting.
>
> “Inpaint Anything” and “Flux Inpainting” are not efficiency-oriented methods; they target general image completion rather than accelerating or scaling diffusion sampling. **These models do not address the performance challenges posed by high-resolution sinograms**, so they are not suitable baselines for evaluating HiSin’s contribution, which centers on efficient diffusion inference.
>
> ### R1-W3: The use of multi-stage inference
>
> In HiSin, the role of multi-stage inference is primarily to provide a practical framework for handling high-resolution sinograms, while the **novel contributions lie in the sinogram-specific mechanisms built on top of it**—namely, frequency-aware patch skipping and structure-adaptive step allocation. These components reshape the sampling process to fit the spectral and geometric properties of sinograms, and are responsible for the efficiency gains demonstrated in our experiments.
>
> ### R1-Q2: Necessity of comparing multiple methods
>
> As dicussed in Section 4.1 line 403 of the main paper, our baselines are grouped into three categories, each serving a distinct purpose. **Diffusion-based methods**(RePaint, DiffIR, HiDiffusion) provide direct comparisons within the same generative backbone as HiSin. **Transformer-based methods** (SinoTx, ICT) represent strong modern inpainting architectures, and among them, **SinoTx is specifically designed for sinogram data**, making it an important domain-relevant reference. Finally, **classical completion approaches** offer traditional baselines commonly used in inpainting baselines. Together, these categories form a coherent and necessary set for evaluating HiSin across the major methodological families.
>
> ### R1–Q3: How baselines are trained and evaluated
>
> “All baselines are retrained from scratch” refers to **training each method on the sinogram datasets used in our experiments**, following the official instructions provided in their respective GitHub repositories. For every baseline, we use the authors’ official codebases and recommended training procedures. All models are trained on the same training split and evaluated on the same test split, ensuring a consistent and equitable comparison across methods.

---

> > ### Comment · Reviewer_dL3A · 2025-11-28
> > **I acknowledge the practical value in resolving the OOM issue and have raised my score to 4. However, I still lean towards rejection as the method is too domain-specific for ICLR.**
> >
> > I thank the authors for their detailed response and clarifications.
> >
> > **On Baselines and Training Details**: I appreciate the clarification regarding the training setup for RePaint. I acknowledge that retraining the diffusion backbone on the specific sinogram domain is necessary for a fair comparison, and I accept the authors' explanation that "retraining" referred to this domain adaptation process. Regarding the choice of baselines, I also accept the justification that general-purpose large models like "Flux" or "Inpaint Anything" may not be suitable comparisons given the paper's specific focus on inference efficiency and memory constraints. The comparison with HiDiffusion (ECCV 2024) is indeed relevant.
> >
> > **On Novelty**: I acknowledge that HiSin has successfully addressed the out-of-memory (OOM) issue that standard baseline models fail to tackle.
> >
> > But the core innovations of the proposed method—specifically the frequency-aware patch skipping and structure-adaptive step allocation—are heavily engineered based on the specific physical characteristics of sinogram data (e.g., spectral sparsity and large sparse backgrounds). While effective for this specific application, these techniques lack generalizability to other data modalities or broader representation learning tasks.
> >
> > In my view, this work is a good application paper that would be better suited for a premier venue focused on medical imaging or computerized tomography (e.g., MICCAI or IEEE TMI), where the domain-specific engineering contributions would be more appropriately weighted. For ICLR, which emphasizes generalizable learning representations, I find the contribution too niche. Therefore, I will raise my rating to 4 (Borderline Reject). However, due to the limited generalizability and scope mismatch with ICLR, I still lean towards rejection.

---

### Author Response · Authors · 2025-12-03
**Author Final Remarks**

We appreciate the reviewers’ time and constructive feedback.

HiSin introduces a novel multi-level inference-time diffusion framework **motivated by reducing computational cost**, including memory usage and inference time in high-resolution inpainting. It combines frequency-aware patch skipping and structure-adaptive step allocation to allocate computation adaptively, enabling diffusion models to scale efficiently without loss of fidelity. Experiments show consistent **reductions in peak memory and runtime while maintaining comparable or better inpainting quality**, confirming the its effectiveness and necessity.

We have carefully addressed all the reviewers' comments. In terms of the **scope of the work**, HiSin is designed for high-resolution structured signals under strict memory and computation budgets, with sinograms—routinely produced in industrial, materials, and medical imaging—representing one of the most demanding and practically important cases of this setting. Besides, we believe HiSin helps fill an existing gap in efficient diffusion inference for large-scale structured domains.

---

### Note · Authors · 2026-01-20

I have read and agree with the venue's withdrawal policy on behalf of myself and my co-authors.